# A Novel Cortisol Immunosensor Based on a Hafnium Oxide/Silicon Structure for Heart Failure Diagnosis

**DOI:** 10.3390/mi13122235

**Published:** 2022-12-16

**Authors:** Hamdi Ben Halima, Nadia Zine, Joan Bausells, Nicole Jaffrezic-Renault, Abdelhamid Errachid

**Affiliations:** 1Institut de Sciences Analytiques (ISA)-UMR 5280, Université Claude Bernard Lyon 1, 5 rue de la Doua, 69100 Lyon, France; 2Institute of Microelectronics of Barcelona (IMB-CNM, CSIC), Campus UAB, Bellaterra, 08193 Barcelona, Spain

**Keywords:** heart failure, cortisol, electrochemical impedance spectroscopy, capacitance, hafnium oxide

## Abstract

Assessing cortisol levels in human bodies has become essential to diagnose heart failure (HF). In this work, we propose a salivary cortisol detection strategy as part of an easily integrable lab-on-a-chip for detection of HF biomarkers. Our developed capacitive immunosensor based on hafnium oxide (HfO_2_)/silicon structure showed good linearity between increasing cortisol concentration and the charge-transfer resistance/capacitance. Moreover, the developed biosensor was demonstrated to be highly selective toward cortisol compared to other HF biomarkers such as tumor necrosis factor (TNF-α) and N-terminal pro-brain natriuretic peptide (NT-proBNP). The precision of our developed biosensor was evaluated, and the difference between the determined cortisol concentration in saliva and its expected one is <18%.

## 1. Introduction

Heart failure (HF) is a cardiovascular chronic disease caused by structural abnormalities of the heart that make it unable to fill or pump out blood, resulting in lower delivery of oxygen. Many different factors such as cardiac abnormalities (e.g., structural abnormalities including stiffness of the ventricular chambers, functional abnormalities including mitral incompetence, etc.), coexisting conditions (e.g., diabetes, anemia, sleep apnea, etc.), or other factors (e.g., age, genetic influence, lifestyle, etc.) can be the cause of HF [1,2].

Cortisol (also called 11β, 17α, 21-trihydroxypregn-4-ene-3,20-dione) is a steroid hormone of the glucocorticoids family, whose chemical formula is C_21_H_30_O_5_ (Figure 1) and whose molar mass is 362.46 g/mol. Cortisol is a crucial glucocorticoid hormone that is produced by the zona fasciculate of the adrenal gland. It is well known as a “stress hormone”, and it takes part in the regulation of various physiological functions including energy metabolism, electrolyte balance, blood pressure, and cognitive function [3]. In addition, it contributes to the homeostasis of the adrenal [4], cardiovascular [5,6], immune [7], and endocrine systems [8]. Moreover, it plays a key role in brain regions that are important for cognitive learning, retrieval, encoding, and memory consolidation [9]. This steroid hormone thus represents a potential biomarker for numerous pathological conditions and diseases [10], as well as a useful clinical indicator for relapse vulnerability in chronic alcohol use.

In humans, cortisol is secreted from the outside of the adrenal cortex of the adrenal gland. It is mainly released in response to stress and a low glucose concentration in the blood (hypoglycemia) [11,12,13]. Stress is a condition that releases glucocorticoids, such as cortisol, and catecholamines, such as adrenaline. These two types of molecules can have effects on memory. Indeed, short exposure improves memory, but prolonged exposure can damage the hippocampal cells, which plays an essential role in memory and space navigation [14,15,16]. Stress can also lead to high production of cortisol and cause an allostatic load, which can result in various modifications in the human body. Among these are, for example, mood disorders, anxiety disorders, diseases, obesity, or even atrophy of the nerve cells of the brain. An increase in reactive oxygen species (ROS) is linked to several serious pathologies [17], for example, rheumatoid arthritis [18], neurodegenerative diseases [19], atherosclerosis [20], and heart diseases such as heart failure [17,21,22]. In addition, oxidative stress is an important pathophysiological pathway in the development and progression of heart failure [21,23,24], making cortisol a potential biomarker for heart failure. Cortisol plays an essential role in oxidative stress, which is the result of an imbalance between the production of reactive (radical) oxygen species (ROS) and the body’s defenses (in terms of antioxidant cellular capacities). Indeed, an increase in cortisol can cause an excess of oxidative stress in the human body and trigger the oxidation of many cells, making them less functional and leading to heart failure in a person [25]. Studies of patients with heart failure have shown that higher cortisol levels indicate a higher risk of death [26,27]. High cortisol levels for a long time can also lead to protein breakdown and muscle loss [28]. At the bone level, cortisol reduces bone formation. It causes osteoporosis in the long term (decreased density of bones), making the bones more fragile, and therefore leads to an increase in bone fracture [29]. Cortisol also has harmful effects on the immune system. In the event of wounds and a high level of stress, an increase in the cortisol level will be observed, and the healing time of the wound will therefore be longer [30]. The amount of cortisol in the blood varies throughout the day, following the circadian cycle. The highest level is early in the morning (around 8 am) and its lowest level from midnight to 4 am [31,32]. In a healthy person, the cortisol concentration can vary from 1 to 12 ng/mL in the morning and from 0.1 to 3 ng/mL in the evening [33,34]. By comparison, a person with heart failure will have greater amounts than a healthy person.

However, the monitoring of cortisol levels must be carried out on the same person because the cortisol level is specific to each, depending on their circadian cycle [32], their diet [35], alcohol consumption [36], and quality of sleep, etc. There are different solutions to regulate the cortisol level in the human body. Thus, some of the factors reducing cortisol levels are, for example, taking magnesium after aerobic exercise [37], taking fish oil [38], music therapy [39], massage therapy [40], laughter and humor [41], and regular dancing [42], etc. Nowadays, cortisol is usually determined by enzyme immunoassay (EIA) [43,44], enzyme-linked immunoassay (ELISA), and radioimmunoassay (RIA) [45], as well as chromatographic techniques coupled with mass spectrometry (MS) or tandem MS/MS [46,47,48]. Commonly, the principal limitations of these steps lie in their high costs, the long run-time, and the requirement of sophisticated technical skills. Herein, a novel and highly selective approach is proposed for detecting cortisol. Similarly, Ben Halima’s approach [49,50,51], a substrate based on hafnium oxide/silicon structure, was biofunctionalized by immobilizing an anti-cortisol antibody onto the ISFET surface after functionalization with 11-triethoxysilyl undecanal (TESUD) by a vapor-phase method in a saturated medium, exploiting the reaction between the aldehyde and the N-terminus of the antibodies. Electrochemical impedance spectroscopy (EIS) was used for analyte detection due to its ability to detect variations in resistance and capacitance induced by binding events, thus enhancing device sensitivity. The linear range, accuracy, precision, and limit of detection (LOD) of the biosensor were investigated to reach a preliminary validation of the device. Selectivity was confirmed by analyzing TNF-α and the gold standard HF biomarker N-terminal pro-brain natriuretic peptide (NT-proBNP) samples [52]. Finally, our cortisol concentration was quantified with our biosensor using the standard addition method (SAM) in real saliva samples. The obtained results prove our biosensor to be a promising tool for cortisol detection in saliva. To the best of our knowledge, this is the first biosensor that uses HfO_2_/silicon structures for cortisol detection. Due to its high K (dielectric constant), HfO_2_ leakage is reduced, and the gate capacitance is enhanced when compared to SiO_2_. This type of structure has already been used for the sensitive detection of interleukin-10 [53]. The use of these materials from microelectronics will allow their integration on a silicon lab-on-a-chip.

## 2. Materials and Methods

### 2.1. Materials and Chemicals

Easy Drop OCA 20 (DataPhysics Instruments (Filderstadt, Germany) was used to perform the contact angle measurement (CAM). The HfO_2_ substrate surface was activated using UV/Ozone Procleaner^TM^ (BioForce, Konstanz, Germany), which created –OH groups on its surface. All experiments were carried out in a Faraday cage at room temperature (20 ± 2 °C). The counter platinum electrode and the reference Ag/AgCl electrode were from BVT Technologies (BVT Technologies, Strážek, Czech Republic). The potentiostat used to carry out the EIS measurements is a VMP3 multichannel (Biologic-EC-Lab Seyssinet-Pariset, France). Data acquisition and modeling were carried out on EC-Lab software (V11.30, BioLogic, Seyssinet-Pariset, France). Phosphate buffer saline solution (PBS) tablets, ethanolamine (purity ≥ 98%), and pure ethanol (purity 95.0%) were purchased from Sigma-Aldrich (Saint-Quentin-Fallavier, France), and 11-triethoxysilyl undecanal (TESUD, 90%) from ABCR (Karlsruhe, Germany). Hydrocortisone (cortisol, purity 99%, Cat. No. ab141250) was from Abcam (Cambridge, UK). Anti-cortisol antibody (Cat. No. XM210) was from Novus Biological (Noyal Châtillon sur Seiche, France). NT-proBNP (Cat. No. 8NT2) was from HyTest (Turku, Finland). Recombinant human TNF-α (Cat. No. 210-TA) was supplied by BioTechne (R&DSystems, noyal chatillon sur seiche, France). Ultrapure water (resistivity > 18 MΩ cm) was produced by the Elga PURELAB Classic system (ELGA LabWater, high Whycombe, UK). PBS tablets were used to create PBS buffer by dissolving it in ultrapure water, thus yielding a 0.01 M phosphate buffer (pH 7.4) with 0.0027 M potassium chloride and 0.137 M sodium chloride, as indicated by the supplier.

### 2.2. Fabrication Technology of HfO_2_/Silicon Structures

A thin-film layer of hafnium oxide (HfO_2_) high-κ dielectric was deposited by atomic layer deposition (ALD) on top of a thin silicon dioxide (SiO_2_) layer grown thermally on silicon wafers. The 4-inch <100> p-type silicon (p-Si) wafers were first cleaned with a 5% of hydrofluoric acid (HF) solution to remove any remaining organic compounds, right before a controlled dry thermal oxidation process at 800 °C to grow a thin interfacial 30 Å SiO_2_ layer. Following the ALD process, a chemical vapor deposition (CVD) technique based on self-terminating gas-substrate reactions was carried out at 200 °C in a Savannah-200 apparatus from Cambridge NanoTech (Cambridge MA, USA), where the wafer surface was exposed to alternating precursor pulses of tetrakis (dimethylamido) hafnium (TDMAH/N_2_) and water vapor (H_2_O/N_2_), forming a single monoatomic layer of HfO_2_ per cycle. Therefore, it was considered that the growth per cycle (GPC) reached a thickness of 20 nm. Finally, a post-deposition annealing process (PDA) was carried out by rapid thermal annealing at different temperatures of 400 °C, 500 °C, and 600 °C, studying the effect on the distribution of interface trap charges and its consequent effect in the dielectric constant by characterizing the sample surface roughness with atomic force microscopy ( Agilent 5500 AFM) (Agilent Technologies, Palo Alto, CA, USA) in tapping mode, as well as characterizing the electrical capacitive behavior. Homogenization of the HfO_2_ surface was obtained at 600 °C, as shown in Figure 2.

### 2.3. Functionalization of HfO_2_ Surface with Antibodies

For the fabrication of the biosensing platform, HfO_2_ was first functionalized using silane aldehyde. For this purpose, bare HfO_2_ substrates were cleaned by sonication in acetone, followed by thorough rinsing in ultrapure water. Surface activation of the HfO_2_ substrates was performed using a UV/Ozone Procleaner^TM^ in order to create –OH groups at the hafnium oxide surface for grafting silane aldehyde. Afterwards, the substrates were thoroughly rinsed and sonicated in ultrapure water. The active HfO_2_ substrates (with –OH) were functionalized using TESUD using the vapor-phase method [50,51,53,54]. Then, the substrates were placed in an oven at 100 °C. After baking, they were rinsed with absolute ethanol and dried with nitrogen. Subsequently, the functionalized substrate surface was incubated with anti-cortisol antibody (10 μg/mL) already diluted in PBS. Finally, the remaining aldehyde groups were blocked by a treatment with ethanolamine (1% *v*/*v*) in PBS buffer. This step is crucial to prevent any nonspecific bonding phenomenon at the detection stage of cortisol (see Figure 3).

### 2.4. Magnetic Nanoparticles Biofunctionalization

The magnetic nanoparticles were prepared with a magnetic core of Fe_3_O_4_ surrounded by styrene/DVB/ACPA polymers that included COOH as the terminal functional group [55]. Initially, they were washed twice with PBS buffer (pH 7.4) and subsequently the COOH group was activated using a mixture of EDC/NHS at 100 mM in PBS. A magnetic field was used to separate the nanoparticles from the storage solution. Anti-cortisol antibody (10 μL) at 100 μg/mL was added to the mixture and incubated with slow stirring at room temperature for 90 min. The nonreacted active carboxylic acid groups were blocked with bovine serum albumin (BSA) (0.1%) in PBS buffer for 30 min. The antibody-coated magnetic nanoparticles were then separated from the mixture, resuspended in 1 mL of PBS buffer, and used for the incubation. The procedure described above is summarized in Figure 4.

### 2.5. Characterization of HfO_2_ Surface Using Contact Angle Measurement (CAM)

The HfO_2_ substrate was characterized using the contact angle measurement system to follow each functionalization step by controlling the surface’s hydrophilicity. The contact angle measurements were performed using ultrapure water. The dosing volume was 5 μL with a dosing rate of 5 μL/s. 

### 2.6. Electrochemical Measurements with HfO_2_/Silicon Structures

Electrochemical impedance spectroscopy (EIS) is an efficient technique for investigating interfacial properties on surface-modified working electrodes. Measurements were made on the HfO_2_/silicon structure with an aluminum backside contact (500 nm thick) as the working electrode, with a counter platinum electrode CE, and an Ag/AgCl reference electrode (Figure 5). All measurements were made with freshly prepared PBS solution that required 8 mL to fill the cell, while the analysis was performed inside a Faraday cage. A VMP3 multichannel potentiostat purchased from Biologic-EC-Lab was used. The preliminary plot established the required potential on the HfO_2_/silicon structure: −1.5 V versus Ag/AgCl, in the accumulation range for silicon. Starting from the lowest, increasing cortisol concentrations were added, incubated for 30 min at 4 °C before impedimetric measurements with the following conditions: EIS frequency ranging from 5 Hz to 100 kHz and a sinus amplitude of 25 mV with polarization potential of −1.5 V-. The scan time was 26 s/scan. Data acquisition and analysis were accomplished using EC-Lab software V11.30. The obtained EIS data were modeled by the EC-Lab software using the Randomize + Simplex method. Here, randomize was fixed at 5000 iterations and the fit fixed at 5000 iterations.

Capacitive measurements were performed at 862 Hz, in a potential range from −0.8 V (accumulation range) to +0.5 V (inversion range).

## 3. Results

### 3.1. Surface Characterization

To assess the effectiveness of the functionalization, contact angle measurements were performed on bare HfO_2_, after activation by UV/Ozone, and after functionalization with TESUD. A contact angle of 53.9 ± 3° revealed a slightly hydrophilic nature on bare HfO_2_, which agrees with Lee’s values [53]. After surface oxidation with UV/ozone activation, the HfO_2_ surface became highly hydrophilic at 12.7 ± 5° due to the high amount of hydroxyl groups on the surface. After the functionalization process with the TESUD, the contact angle increased again to 80.3 ± 2°. This hydrophobic character can be explained by the presence of TESUD hydrocarbon chains (see Figure 6).

### 3.2. Biosensor Calibration by EIS Measurements

Bare HfO_2_ substrates were cleaned and activated with TESUD, as previously described. The anti-cortisol antibody was immobilized in a conventional three-electrode cell overnight at 4 °C, then rinsed with PBS. Following the immobilization of anti-cortisol antibody onto the substrate, the antibody-modified HfO_2_/silicon structure was incubated with cortisol at different increasing concentrations. 

Impedance results were presented as a Nyquist plot, as shown in Figure 7A. 

An attempt at signal enhancement was made by an additional incubation of the electrode with the MNPs functionalized with the complementary antibody. An SEM image, obtained using FEI Quanta FEG 250 (Thermo Fisher Scientific Inc, Waltham, MA USA) (Figure 8), shows a homogenous distribution of MNPs on the functionalized HfO_2_ surface. Nevertheless, the effect of the enhancement of the signal was very disappointing.

The Nyquist plots of HfO_2_ modified with anti-cortisol antibody, followed by increasing cortisol concentrations, were fitted with a Randles equivalent circuit (insert in Figure 7A). Here, the components are as follows: Rs corresponds to the resistance of the electrolyte solution; C_dl_ is the double-layer capacitance that is in parallel with R_ct_, which is the charge-transfer resistance; and Zw is the Warburg impedance. The values of the extracted parameters are presented in Table 1.

An increase in R_ct_ (charge-transfer resistance) can be observed from R_0_ (no cortisol) at 206 kΩ to 302 kΩ for a cortisol concentration of 2 ng/mL. The change in R_ct_ demonstrates the biorecognition of the cortisol by the anti-cortisol antibody grafted on the HfO_2_ surface. The R_ct_ increased: 370 kΩ for 10 ng/mL; 450 kΩ for 15 ng/mL; 620 kΩ for 50 ng/mL. The ∼104 kΩ variations between the anti-cortisol antibody modified electrode and the first concentration of cortisol at 2 ng/mL demonstrate that the quantification limit of the immunosensor is 2 ng/mL. In Figure 7B (black), the relative variation of Rct versus the cortisol concentration plot produced a linear relationship ranging from 2 ng/mL to 50 ng/mL, with R^2^ = 0.9722 with a slope of 0.03 (ng/mL)^−1^. A cross-selectivity study was performed to assess the level of nonspecific binding. For this purpose, two other HF biomarkers were used, namely TNF-α and NT-proBNP, using the same conditions and concentrations as for cortisol. From Figure 7B, it can be seen that the biosensor is demonstrated to be highly selective towards cortisol when compared to both TNF-α and NT-proBNP. The sensitivity for TNF-α (green) is 0.0005 (ng/mL)^−1^, and for NT-proBNP (Red) the sensitivity is 0.0026 (ng/mL)^−1^. The immunosensor is 200 times more sensitive to cortisol than TNF-α, and 12 times more sensitive to cortisol than NT-proBNP.

### 3.3. Biosensor Calibration by Capacitive Measurements

Additionally, capacitive measurements were performed to characterize the semi-conducting behavior of HfO_2_/silicon structures for each cortisol concentration. The biosensor was maintained in the electrochemical cell and incubated in PBS containing cortisol at different concentrations. The biosensor was then rinsed with PBS to remove any adsorbed proteins and analyzed afterwards by capacitance measurements using PBS as an electrolyte. This procedure of immunosensor incubation was carried out for all cortisol concentrations (2 ng/mL to 50 ng/mL). The detection of cortisol at various concentrations is shown in Figure 9A. Here, the Cs/Cmax–voltage curves show a shift in the positive direction with increasing cortisol concentrations, which confirms a flat-band voltage variation due to the increase in electrical charge at the surface of the HfO_2_/silicon structure. 

Figure 9B shows the calibration curve of the HfO_2_-based capacitive immunosensor in a linear range from 2 ng/mL to 50 ng/mL of cortisol. The calibration curve has been presented as the absolute value of potential shift |E-E_0_| as a function of cortisol concentration, where E is the potential for different cortisol concentrations and E_0_ is the potential with no cortisol in the solution. The slope of the straight line is 0.0023 (ng/mL)^−1^.

The specificity of the biosensor prepared was studied using other biomarkers of heart failure, such as TNF-α and NT-proBNP. Capacitance measurements were performed using the same experimental process (Figure 10A,B). TNF-α and NT-proBNP were detected in the same linear range from 2 to 50 ng/mL with sensitivities of 0.0001 (ng/mL)^−1^ and 0.000001 (ng/mL)^−1^, respectively. In this configuration of measurements, the immunosensor is 23 times more sensitive to cortisol than TNF-α, and 2300 times more sensitive to cortisol than NT-proBNP.

The analytical performance of the HfO_2_/silicon-based immunosensor was compared to those of the published electrochemical immunosensors for the detection of cortisol (Table 2). The detection limit of the HfO_2_/silicon-based immunosensor is in the lower range, and its detection range falls in the range of concentrations of cortisol for a healthy person and for a person with HF.

### 3.4. Cortisol Quantification in Saliva Samples by Standard Addition Method (SAM)

To simulate the saliva sample analysis with an unknown cortisol concentration, three aliquots (450 μL) of the pooled saliva sample (PSS) were spiked with different volumes of 100 ng mL^−1^ cortisol standard solution, obtaining a final concentration of 2, 5, and 7 ng mL^−1^ of cortisol, and then named “unknown sample” A, B, and C. SAM was carried out by first preparing SAM samples, where a constant volume (50 μL) of the “unknown sample” was added to each of a quartet of 1.5 mL Eppendorf Lo-bind centrifuge tubes (Sigma-Aldrich, Saint-Quentin-Fallavier, France). A total of 950 μL of PBS was added to the first tube up to obtain sample C0. Then, an increasing volume of the 100 ng mL^−1^ cortisol standard solution was added to each subsequent tube, before rounding the solutions in each tube to 1 mL with PBS, thus obtaining three samples, C1, C2, and C3, with known SAM concentrations, where they were analyzed by EIS.

Data obtained from SAM analysis are summarized in Table 3. The precision of our method was calculated by estimating the difference between the cortisol concentrations determined using SAM and their expected values. Based on these results, our biosensor shows good precision to determine the unknown concentration of cortisol in real saliva. These results confirmed that our biosensor was highly sensitive to the slight variation of the unknown concentration of cortisol. 

## 4. Conclusions

In this study, we present the development of a label-free, highly sensitive, accurate, fast biosensor using EIS/capacitive measurement for cortisol detection in PBS/saliva while also showing good selectivity toward cortisol in the presence of other HF biomarkers (TNF-α and NT-proBNP). Tests performed in PBS showed a linear relationship with the increase in cortisol concentration and the resistance/capacitance of our transistor (R^2^ was always > 0.97), showcasing the capability of our biosensor to quantitatively detect cortisol with a detection limit of 0.66 ng.mL^−1^ and a dynamic range from 2 to 50 ng.mL^−1^. In addition, our biosensor showed high precision for detecting cortisol in unknown samples using the standard addition method and high sensitivity to small variations of cortisol concentration. This biosensor thus represents a promising bioanalytical tool for accurate quantification of cortisol in saliva to monitor symptoms of inflammation in HF patients. Moreover, the use of materials from microelectronics will allow their integration into silicon lab-on-a-chip devices.

## Figures and Tables

**Figure 1 micromachines-13-02235-f001:**
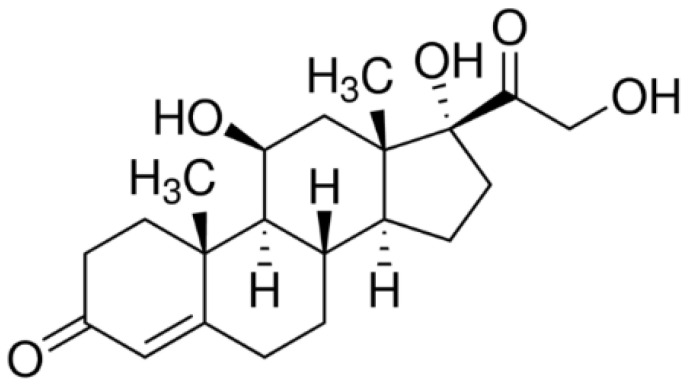
Semi-developed cortisol formula.

**Figure 2 micromachines-13-02235-f002:**

Characterization of the sample surface roughness with atomic force microscopy (AFM) in tapping mode using post-deposition annealing process (PDA) at different temperatures of 400 °C (**a**), 500 °C (**b**), and 600 °C (**c**).

**Figure 3 micromachines-13-02235-f003:**
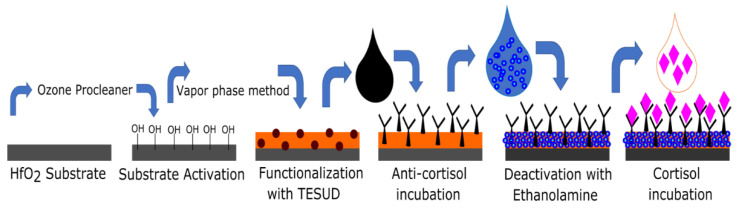
Functionalization procedure of the HfO_2_ surface.

**Figure 4 micromachines-13-02235-f004:**
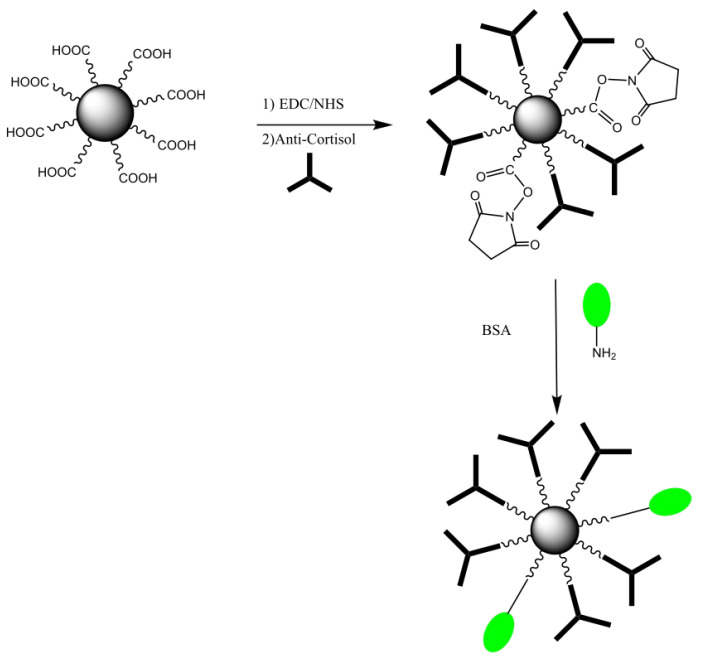
MNPs-COOH biofunctionalization with anti-cortisol antibody.

**Figure 5 micromachines-13-02235-f005:**
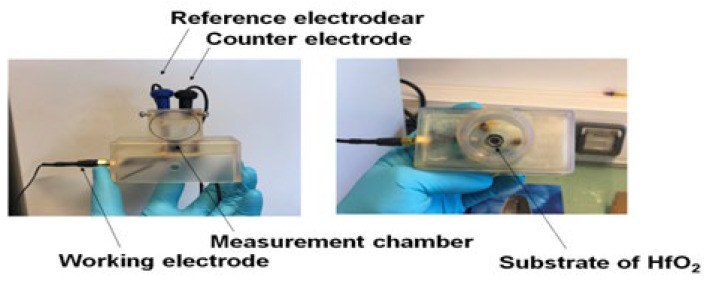
Experimental setup.

**Figure 6 micromachines-13-02235-f006:**
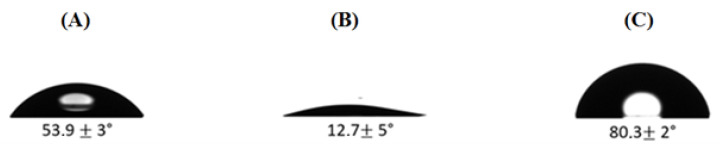
Contact angle on (**A**) bare HfO_2_ before surface activation; (**B**) bare HfO2 after surface activation by UV/ozone; (**C**) following TESUD formation.

**Figure 7 micromachines-13-02235-f007:**
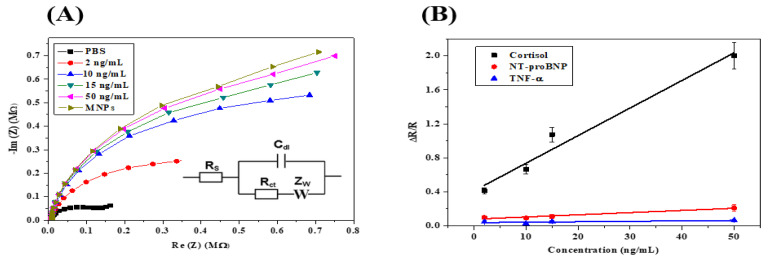
(**A**) Example of Nyquist plots for the Randles equivalent circuit model obtained by analyzing cortisol standard solutions in PBS (2, 10, 15, and 50 ng mL^−1^). EIS frequency ranged from 5 Hz to 100 kHz, and a sinus amplitude of 25 mV with polarization potential of −1.5 V; (**B**) sensitivity curves obtained by analyzing standard solution containing cortisol or other HF biomarkers (e.g., TNF-α and NT-proBNP) in the concentration range 2–50 ng mL^−1^ using the HfO_2_ substrate functionalized with anti-cortisol antibody.

**Figure 8 micromachines-13-02235-f008:**
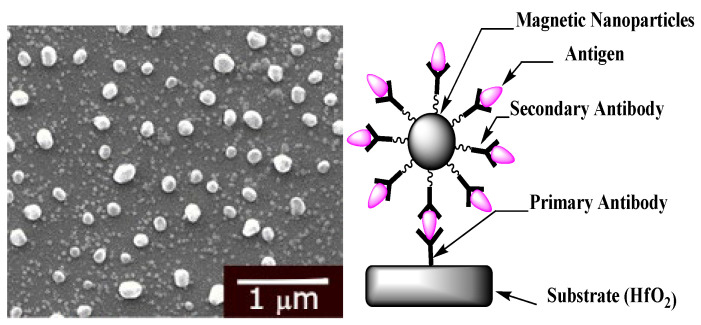
SEM image of functionalized TESUD substrates with Ab-Ag-Ab-MNP (magnetic nanoparticles) biorecognition: left side: SEM image; right side: scheme.

**Figure 9 micromachines-13-02235-f009:**
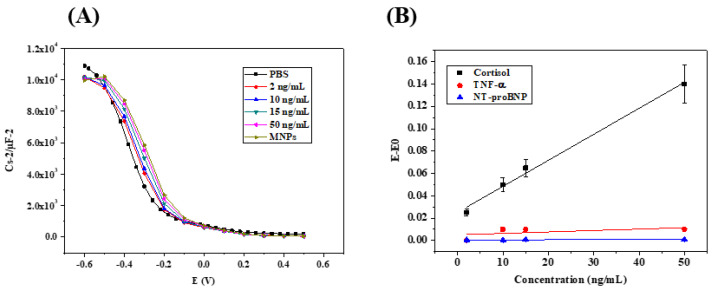
(**A**) Capacitance–voltage plots for cortisol detection using the capacitance biosensor; (**B**) the calibration curves of the cortisol detection (black curve) and the two interferences: TNF-α (red curve) and NT-proBNP (blue curve).

**Figure 10 micromachines-13-02235-f010:**
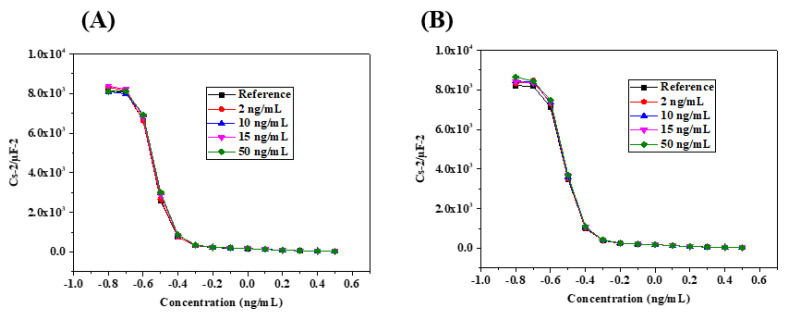
(**A**) Capacitance–voltage plots for TNF-α detection; (**B**) for NT-proBNP detection using the capacitive immunosensor.

**Table 1 micromachines-13-02235-t001:** Fitting parameters obtained from the equivalent circuit of the cortisol-based HfO_2_ immunosensor.

Cortisol Concentration (ng mL^−1^)	Rs (Ω)	Q_2_ (nF.s ^(a−1)^)	R_ct_ (MΩ)	s2 (kΩ.s^−1/2^)	*χ*2
0	9181	10.91	0.206	0.547	1.754 × 10^−3^
2	27,031	10.69	0.302	1.88	0.514 × 10^−3^
10	29,128	10.87	0.370	2.925	1.845 × 10^−3^
15	33,652	10.84	0.450	4.419	1.050 × 10^−3^
50	44,199	12.3	0.620	3.633	1.436 × 10^−3^

**Table 2 micromachines-13-02235-t002:** Comparison of the analytical performance of the HfO_2_/silicon-based immunosensor to those of the published electrochemical immunosensors for the detection of cortisol.

Technique	Electrode	Immobilizing Biomolecules	Analyte	Linear Range (ng mL^−1^)	Limit of Detection (LOD) (ng mL^−1^)	References
EIS	-	Anti-cortisol antibody	Human Tears	0.05–200	21.66	[56]
SWV	Graphite	Anti-cortisol antibody	Human saliva	0.5–55.5	1.7	[57]
Amperometry	Reduced graphene oxide	Anti-cortisol antibody	Human saliva and sweat	0.1–200	0.1	[58]
EIS	Au	Anti-cortisol antibody + BSA	Fish plasma	1440–2170	2750	[59]
EIS	Palladium + MoS_2_	Anti-cortisol antibody	Human sweat	1–500	1	[60]
EIS/capacitance	HfO_2_	Anti-cortisol antibody	Real saliva	2–50	0.66	This work

**Table 3 micromachines-13-02235-t003:** Data obtained from the analysis of three saliva samples (corresponding to three aliquots of PSS spiked with different cortisol) by SAM.

Sample Name	Added Concentration (ng/mL)	Dilution Factor	Calculated Concentration	Bias
Sample A	2	19.9	2.4 ± 0.3 ng mL^−1^ (CV = 1%)	14%
Sample B	5	19.5	4.7 ± 0.7 ng mL^−1^ (CV = 5%)	18%
Sample C	7	19.7	7.1 ± 0.2 ng mL^−1^ (CV = 1%)	15%

## Data Availability

The data that support the findings of this study are available from the corresponding author upon reasonable request.

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
