# Peer review of "A Novel Cortisol Immunosensor Based on a Hafnium Oxide/Silicon Structure for Heart Failure Diagnosis"

_micromachines, 2022, doi:10.3390/mi13122235_

Round 1

Reviewer 1 Report

Review Report

Manuscript ID:  micromachines-2098120

Title

A novel cortisol immunosensor based on a hafnium ox-ide/silicon structure for heart failure diagnosis

Comments and suggestions

This work regarding, to the development of relatively cheap and more efficient biosensor by using HfO2/silicon materials for detecting cortisol level. This idea is quite good for heart related diseases studies, but it has lack of novelty through out the manuscript and extensive English corrections are required. I strongly recommend, this manuscript will go under major revisions   

1.               Line no. 93, The authors claims “this is the first biosensor that uses HfO2/silicon”, if its true show me the evidence, refer the following articles and you may discusses in introduction section.

https://doi.org/10.1016/j.talanta.2022.123802, https://doi.org/10.1002/elan.202200141, https://doi.org/10.1016/j.compositesb.2021.109231,  

2.               Line no. 120, try to use traditional format of “4-inch<100>p-type silicon”.

3.               Line 121, “5% HF” what is the meaning this, you have declared HF is heart failure in abstract. But here ?, You must understand the significant of the short forms.  

4.               Figure 2 is blurry,  You must improve the quality of the figure 2.

5.               Line 144, why are write” ultrapure water (Millipore Milli-Q)”, that should be present in the first paragraph of materials and methods sections. Try to stick in one term, ie, ultra or Millipore.

6.               Line no 148, 168, 303, what you meant by “TESUD”,”BSA”,
“LOD”, you should mention the abbreviations before use.

7.               How many table no. 1 in your manuscript, you must revise it.

8.               Figure 6, you must re draw graphs with error bars; thick scale bar and you may increase the font size.  

9.               Figure 6, how many times you have repeated this experiment, what is the standard deviation.

10.            Line no 246, you may modified “Cortisol concentration (ng mL−1)”, instead of “Cortisol concentration”.

11.            Figure 8,9, must re draw with greater font size with error bar. Need to improve the quality.

12.            Table 2 should re format and change borders.

13.            The abstract and conclusion should be rewrite again, the author must understand what is abstract? “it’s the compressed version of the whole article”. What is conclusion? “it’s the compressed version of the Results and discussions”

14.            You must include recent references in reference section

Author Response

Reviewer 1

Comments and suggestions

This work regarding, to the development of relatively cheap and more efficient biosensor by using HfO2/silicon materials for detecting cortisol level. This idea is quite good for heart related diseases studies, but it has lack of novelty through out the manuscript and extensive English corrections are required. I strongly recommend, this manuscript will go under major revisions   

We thank the reviewer for his fruitful comments and are grateful for his judgment on our article.
After revision, English was revised by a native English person.

  1. Line no. 93, The authors claims “this is the first biosensor that uses HfO2/silicon”, if its true show me the evidence, refer the following articles and you may discusses in introduction section.

https://doi.org/10.1016/j.talanta.2022.123802, https://doi.org/10.1002/elan.202200141, https://doi.org/10.1016/j.compositesb.2021.109231,  

We thank the reviewer for his comments, however, the two articles (https://doi.org/10.1016/j.talanta.2022.123802; https://doi.org/10.1002/elan.202200141)  mentioned above are based on Silicon nitride and not the HfO2; these two articles were added in the introduction section. Concerning, the third article (https://doi.org/10.1016/j.compositesb.2021.109231) which is based on HfO2, it is devoted the detection of hunger hormone ghrelin and not cortisol.

  1. Line no. 120, try to use traditional format of “4-inch<100>p-type silicon”.

The format was modified as follows: 4-inch <100> p-type silicon (p-Si) wafers

  1. Line 121, “5% HF” what is the meaning this, you have declared HF is heart failure in abstract. But here ?, You must understand the significant of the short forms.

We thank the reviewer for his comments, we mentioned in the manuscript the significance of 5% HF which is 5% hydrofluoric acid.

  1. Figure 2 is blurry,  You must improve the quality of the figure 2.

We thank the reviewer for his comments, the quality of figure 2 was improved as much as possible.

  1. Line 144, why are write” ultrapure water (Millipore Milli-Q)”, that should be present in the first paragraph of materials and methods sections. Try to stick in one term, ie, ultra or Millipore.

We thank the reviewer for his comments, we remove (Millipore Milli-Q) in line 144 and we already mentioned “ultra pure water” in line 129, in the materials and method section. 

  1. Line no 148, 168, 303, what you meant by “TESUD”,”BSA”,
    “LOD”, you should mention the abbreviations before use.

We thank the reviewer for his comments, all the significance of these abbreviations was mentioned in the manuscript when they were used for the first time.

TESUD: 11-triethoxysilyl undecanal (lines 99 and 125)

BSA : bovine serum albumin

LOD: Limit of Detection (line 104)

  1. How many table no. 1 in your manuscript, you must revise it.

We thank the reviewer for his comments, the manuscript includes 3 tables we renumbered them.

  1. Figure 6, you must re draw graphs with error bars; thick scale bar and you may increase the font size.

We thank the reviewer for his comments, we re-drawn new figure 7B with error bars and we increased the font size.

  1. Figure 6, how many times you have repeated this experiment, what is the standard deviation.

We thank the reviewer for his comments, we repeated this experiment five time, and the standard deviation for the detection of cortisol is 8%.

  1. Line no 246, you may modified “Cortisol concentration (ng mL−1)”, instead of “Cortisol concentration”.

We thank the reviewer for his comments, in Table 1, first raw, we modified “Cortisol concentration” by Cortisol concentration (ng mL−1)” and it was highlighted in yellow.

  1. Figure 8,9, must re draw with greater font size with error bar. Need to improve the quality.

We thank the reviewer for his comments, we redrawn new figures 9, 10 with greater font size with error bars for figure 8, and we tried to improve the quality as much as possible.

  1. Table 2 should re format and change borders.

In Table 2, borders were changed

  1. The abstract and conclusion should be rewrite again, the author must understand what is abstract? “it’s the compressed version of the whole article”. What is conclusion? “it’s the compressed version of the Results and discussions”

We thank the reviewer for his comments.

The abstract was modified as follows:

Assessing cortisol levels in human bodies has become essential to diagnose heart failure (HF). In this work, we propose a salivary cortisol detection strategy as part of an easily integrable Lab-on-Chip for detection HF biomarkers. Our developed capacitive immunosensor based on hafnium oxide (HfO2)/silicon structure showed good linearity between increasing cortisol concentration and the charge transfer resistance/capacitance. Moreover, the developed biosensor was demonstrated to be highly selective toward cortisol compared to other HF biomarkers such as Tumor necrosis factor (TNF-α) and N-terminal pro-brain natriuretic peptide (NT-proBNP). The precision of our developed biosensor was evaluated, and the difference between the determined cortisol concentration in saliva and its expected one is < 18%.

The conclusion was modified as follows:

In this study, we present the development of a label-free, highly sensitive, accurate, fast biosensor using EIS/capacitive measurement for cortisol detection in PBS/saliva while also showing good selectivity toward cortisol in the presence of other HF biomarkers (TNF-α and NT-proBNP). Tests performed in PBS showed a linear relationship with the increase in cortisol concentration and the resistance/capacitance of our transistor (R2 was always > 0.97), showcasing the capability of our biosensor to quantitatively detect cortisol with a detection limit of 0.66 ng.mL-1 and a dynamic range of 2 to 50 ng.mL-1. In addition, our biosensor showed high precision for detecting cortisol in unknown samples using the standard addition method and high sensitivity to small variations of cortisol concentration. This biosensor thus represents a promising bioanalytical tool for accurate quantification of cortisol in saliva to monitor symptoms of inflammation in HF patients. Moreover, the use of materials from microelectronics will allow their integration in silicon lab-on-a-chip.

  1. You must include recent references in reference section

11 recent references replaced old ones

Reviewer 2 Report

The abstract should be corrected. The following should be emphasized in the abstract: objective, idea, description of idea, methods, results, quantitative comparison of results with significant findings, conclusions.

At the end of the Introduction section, emphasize the scientific contribution of your research.

The conclusion section should be rewritten. Highlight your scientific contribution. Highlight the benefits of your research. Define shortcomings and future research.

Include experimental setup picture.

Justify materials selection.

What about deployment of this research?

All measurements which are carried out should be showcased with pictures.

Results section looks more like a report and not a research paper.

Author Response

We thank the reviewer for his comments and are grateful for his judgment on our article.

The abstract should be corrected. The following should be emphasized in the abstract: objective, idea, description of idea, methods, results, quantitative comparison of results with significant findings, conclusions.

We thank the reviewer for his comments, we change the abstract as requested.

Assessing cortisol levels in human bodies has become essential to diagnose heart failure (HF). In this work, we propose a salivary cortisol detection strategy as part of an easily integrable Lab-on-Chip for detection HF biomarkers. Our developed capacitive immunosensor based on hafnium oxide (HfO2)/silicon structure showed good linearity between increasing cortisol concentration and the charge transfer resistance/capacitance. Moreover, the developed biosensor was demonstrated to be highly selective toward cortisol compared to other HF biomarkers such as Tumor necrosis factor (TNF-α) and N-terminal pro-brain natriuretic peptide (NT-proBNP). The precision of our developed biosensor was evaluated, and the difference between the determined cortisol concentration in saliva and its expected one is < 18%.

At the end of the Introduction section, emphasize the scientific contribution of your research.

We thank the reviewer for his comments, we added a part in order to emphasize the scientific contribution of our research.

To reach this aim, similarly to what has been done by Ben Halima et al. [48–50], substrate based on Hafnium oxide/silicon structure were biofunctionalized by immobilizing anti-cortisol antibody onto the ISFET surface after functionalization with 11-triethoxysilyl undecanal (TESUD) by a vapor-phase method in a saturated medium exploiting the reaction between the aldehyde and the N-terminus of the antibodies. Electrochemical impedance spectroscopy (EIS) was used for analyte detection due to its ability to detect variations in resistance and capacitance upon binding events enhancing device sensitivity. Linear range, accuracy, precision, and limit of detection (LOD) of the biosensor were investigated to reach a preliminary validation of the device. Selectivity was confirmed by analysing TNF-α, and the gold standard HF biomarker N-terminal pro-brain natriuretic peptide (NT-proBNP) samples [51]. Finally, our cortisol concentration was quantified with our biosensor using the standard addition method (SAM) in real saliva samples. The obtained results proved our biosensor as a promising tool for cortisol detection in saliva. To the best of our knowledge, this is the first biosensor that uses HfO2/silicon structures for cortisol detection. Due to its high K (dielectric constant) HfO2 allows the reduction of leakage and the enhancement of the gate capacitance when compared to SiO2. This type of structure was already used for the sensitive detection of interleukin-10 [48]. The use of these materials from microelectronics will allow their integration in silicon labo-on-a chip.

The conclusion section should be rewritten. Highlight your scientific contribution. Highlight the benefits of your research. Define shortcomings and future research.

We thank the reviewer for his comments, we change the conclusion as requested.

In this study, we present the development of a label-free, highly sensitive, accurate, fast biosensor using EIS/capacitive measurement for cortisol detection in PBS/saliva while also showing good selectivity toward cortisol in the presence of other HF biomarkers (TNF-α and NT-proBNP). Tests performed in PBS showed a linear relationship with the increase in cortisol concentration and the resistance/capacitance of our transistor (R2 was always > 0.97), showcasing the capability of our biosensor to quantitatively detect cortisol with a detection limit of 0.66 ng.mL-1 and a dynamic range of 2 to 50 ng.mL-1. In addition, our biosensor showed high precision for detecting cortisol in unknown samples using the standard addition method and high sensitivity to small variations of cortisol concentration. This biosensor thus represents a promising bioanalytical tool for accurate quantification of cortisol in saliva to monitor symptoms of inflammation in HF patients. Moreover, the use of materials from microelectronics will allow their integration in silicon lab-on-a-chip.

Include experimental setup picture.

We thank the reviewer for his comments, we added an experimental setup picture.

Justify materials selection.

What about deployment of this research?

The materials used were justified at the end of the introduction as well as the deployment of this research:

To the best of our knowledge, this is the first biosensor that uses HfO2/silicon structures for cortisol detection. Due to its high K (dielectric constant) HfO2 allows the reduction of leakage and the enhancement of the gate capacitance when compared to SiO2. This type of structure was already used for the sensitive detection of interleukin-10 [48]. The use of these materials from microelectronics will allow their integration in silicon lab-on-a-chip.

All measurements which are carried out should be showcased with pictures.

An experimental set-up picture was added

Results section looks more like a report and not a research paper.

Results section shows the analytical performance of the HfO2/silicon based cortisol immunosensor, compare it the published electrochemical cortisol sensors and applied it to the detection of cortisol in spiked artificial saliva samples.

Reviewer 3 Report

The article micromachines-2098120 entitled “A novel cortisol immunosensor based on a hafnium oxide/sili-2 con structure for heart failure diagnosis” described an electrochemical impedance inmunosensor for cortisol determination based on hafnium oxide (HfO2) surface functionalized with TESUD ((11-triethoxysilyl) undecanal). The anti-cortisol antibody was immobilized on the surface thanks to the reaction of amino groups of the antibody with aldehyde groups generated. Magnetic nanoparticles with carboxylic groups have been used as detection biomaterial, being modified with anti-cortisol antibody. The experimental set-up is well design, and results obtained are good enough. The electrochemical platform has been characterized by AFM measurements and the EIS measurements clearly showed the biosensor respond to different concentrations of cortisol. The developed biosensor has been tested against potential interferences and has been applied in real saliva sample. I consider the work has enough quality and the results are well supported. I recommended publication of this work in Micromachines.

Just to errors detected:

Figure 5 caption subscripts should be corrected (HfO2).

Line 332- Table 1k. should be corrected to Table1.

Author Response

Reviewer 3

The article micromachines-2098120 entitled “A novel cortisol immunosensor based on a hafnium oxide/sili-2 con structure for heart failure diagnosis” described an electrochemical impedance inmunosensor for cortisol determination based on hafnium oxide (HfO2) surface functionalized with TESUD ((11-triethoxysilyl) undecanal). The anti-cortisol antibody was immobilized on the surface thanks to the reaction of amino groups of the antibody with aldehyde groups generated. Magnetic nanoparticles with carboxylic groups have been used as detection biomaterial, being modified with anti-cortisol antibody. The experimental set-up is well design, and results obtained are good enough. The electrochemical platform has been characterized by AFM measurements and the EIS measurements clearly showed the biosensor respond to different concentrations of cortisol. The developed biosensor has been tested against potential interferences and has been applied in real saliva sample. I consider the work has enough quality and the results are well supported. I recommended publication of this work in Micromachines.

Just to errors detected:

Figure 5 caption subscripts should be corrected (HfO2).

Caption of new Figure 6 was corrected as follows:

Figure 6. Contact angle on (A) bare HfO2 before surface activation; (B) bare HfO2 after surface activation by UV/Ozone; (C) following TESUD formation.

Line 332- Table 1k. should be corrected to Table1.

This point was corrected.

Round 2

Reviewer 1 Report

The current format of this manuscript is much improved and need to check spell, type errors by the author.

Reviewer 2 Report

The authors have addressed my comments.